# GlobalBench: A Benchmark for Global Progress in Natural Language Processing

**Yueqi Song[1], Catherine Cui[2], Simran Khanuja[1], Pengfei Liu[3], Fahim Faisal[4],**
**Alissa Ostapenko[1], Genta Indra Winata[5], Alham Fikri Aji[6], Samuel Cahyawijaya[5],**
**Yulia Tsvetkov[1,7], Antonios Anastasopoulos[4], Graham Neubig[1]**

[1]Carnegie Mellon University   [2]Harvard University   [3]Shanghai Jiaotong University
[4]George Mason University   [5]The Hong Kong University of Science and Technology
[6]MBZUAI   [7]University of Washington

## Abstract

Despite the major advances in NLP, significant disparities in NLP system performance across languages still exist. Arguably, these are due to uneven resource allocation and sub-optimal incentives to work on less resourced languages. To track and further incentivize the global development of equitable language technology, we introduce GlobalBench. Prior multilingual benchmarks are static and have focused on a limited number of tasks and languages. In contrast, GlobalBench is an ever-expanding collection that aims to dynamically track progress on *all* NLP datasets in *all* languages. Rather than solely measuring accuracy, GlobalBench also tracks the estimated *per-speaker utility* and *equity* of technology across all languages, providing a multi-faceted view of how language technology is serving people of the world. Furthermore, GlobalBench is designed to identify the most under-served languages, and rewards research efforts directed towards those languages. At present, the most under-served languages are the ones with a relatively high population, but nonetheless overlooked by composite multilingual benchmarks (like Punjabi, Portuguese, and Wu Chinese). Currently, GlobalBench covers 966 datasets in 190 languages and includes 1,128 system submissions spanning 62 languages.[1]

## 1 Introduction

Advances in multilingual natural language processing (NLP) technologies (Dabre et al., 2020; Hedderich et al., 2021) have raised the enticing possibilities of NLP systems that benefit all people around the world. However, at the same time, studies into the state of multilingual NLP have demonstrated stark differences in the amount of resources available (Joshi et al., 2020; Yu et al., 2022) and performance of existing NLP systems (Blasi et al., 2022; Khanuja et al., 2023; Ahia et al., 2023).

Why do these disparities exist? The causes of these disparities are multifarious, but Blasi et al. (2022) argue that one major factor is a problem of incentives and resource allocation. For instance, languages associated with larger economic might

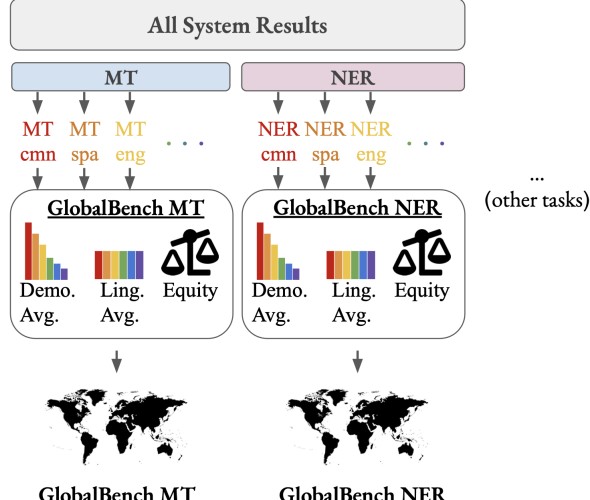

Figure 1: *GlobalBench Design*: A leaderboard for each task is separately maintained. Each leaderboard contains a multi-faceted evaluation of submitted systems, along with a ranking of the most under-served languages. More details can be found in Section 2.

(as measured by GDP of the countries where they are spoken) see more research and resource development, leading to more performant systems.

In this paper, we propose GlobalBench, a new benchmark and leaderboard that is designed to *specifically incentivize the global development of equitable language technologies* that serve speakers of all languages throughout the world. GlobalBench follows the footsteps of other successful multilingual benchmarks such as XTREME (Hu et al., 2020) and XGLUE (Liang et al., 2020), which aggregate results of systems across several tasks to provide a general idea of progress being made in the field of multilingual NLP. However, these benchmarks, by design, are static and lack the goal to be an all-inclusive, ever-expanding collection of datasets. Additionally, they mainly focus on average accuracy over all languages in the dataset, and thus say little about the downstream *utility* and

---

[1]GlobalBench is available at https://github.com/neulab/globalbench.

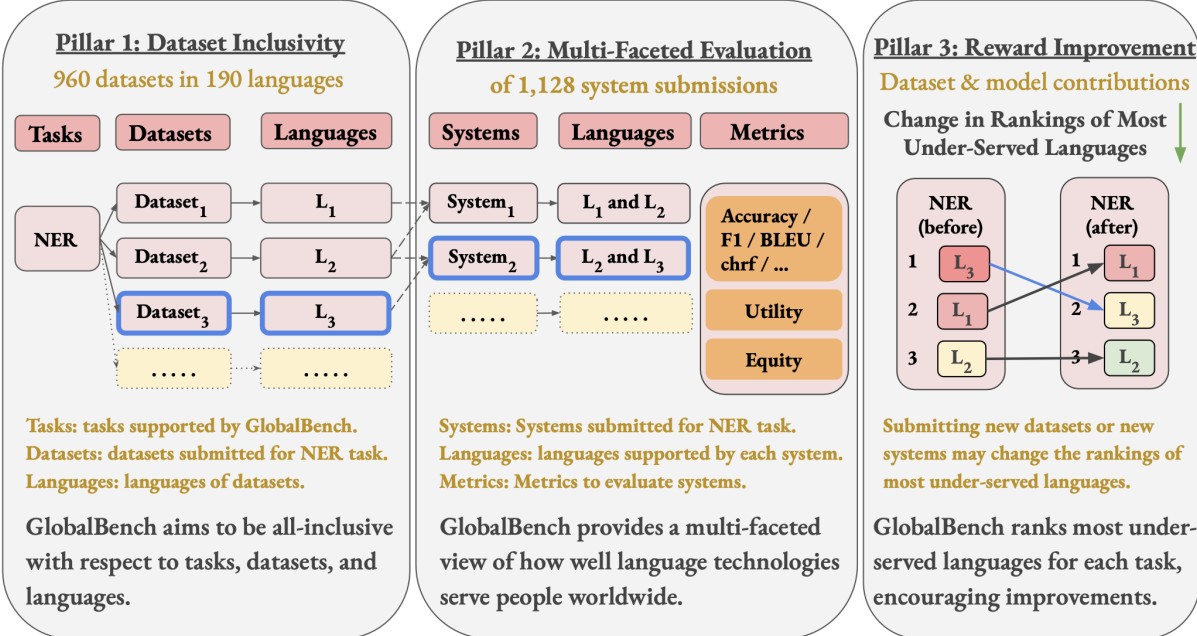

Figure 2: *GlobalBench's Philosophy*: First, we aim to inclusively gather datasets for all tasks and languages. Second, we present a multi-faceted evaluation of systems, going beyond average accuracies across languages, to keep track of the utility and equity of these systems. Third, GlobalBench ranks the most to least under-served languages on the leaderboard of each task, and rewards improvement in utility, which can be achieved through both dataset and model contributions. The lower the ranking of a language the better since a lower ranking indicates it is less under-served. In the above example, the addition of $Dataset_3$ improves the measured utility of $L_3$ in the NER leaderboard, and the addition of $System_2$ improves the measured utility of $L_2$ and $L_3$ for the NER leaderboard.

*equity* of submitted systems across languages.

Hence, in designing GlobalBench, we make a number of intentional design decisions to explicitly promote the improvement of language technology for all of the world's citizens:

- **Inclusive Dataset Selection**: We aim to be able to evaluate *all* datasets in *all* languages for *all* tasks, making it possible to (in theory) cover any language for which a dataset exists.

- **Multi-Faceted Evaluation**: As shown in Figure 1, GlobalBench explicitly considers per-speaker *utility* and *equity* (§2), measuring how close NLP systems have come to equitably covering all speakers in the world, instead of just those in our existing data.

- **Reward Data and Model Contributions**: GlobalBench encourages *improvements* in the state-of-the-art (instead of just measuring the state-of-the-art itself), by identifying under-served languages and rewarding progress on them, both in terms of dataset and model contributions.

In the remainder of this paper, we detail GlobalBench's design principles (§2), how interested research community members can participate (§3), the currently covered tasks and systems (§4), analysis of the current state of NLP viewed through the lens of GlobalBench (§5), related work (§6) and our expectations for the path forward (§7).

All in all, we believe that improving the quality and equity of language technologies for all speakers in the world is one of the paramount challenges of NLP today. In the famous mantra from Peter Drucker, *what you cannot measure, you cannot improve*; GlobalBench is a first step in this direction.

## 2 GlobalBench Design Principles

**Philosophy**: A working example of our guiding philosophy is shown in Figure 2. Our unique reward system incentivizes model builders to not only improve system performance, but also build datasets for new languages. To illustrate the former, let's assume that there were NER datasets for $L_2$ and $L_3$ on GlobalBench, and now researchers build a new system for NER ($System_2$) which is state-of-the-art for languages $L_2$ and $L_3$. This

increases utility for both languages (Equation 3), which is attributed to $System_2$ on our NER leaderboard. For the latter, let's change our assumption to that there was no NER dataset for $L_3$ before, but a pre-trained model ($System_2$) supports this language. Hence, the introduction of $Dataset_3$ helps realize a sharp improvement in utility for $L_3$, which was previously immeasurable (hence, for all practical purposes, zero). Thus, the increase in utility for $L_3$ on the NER leaderboard is attributed to $Dataset_3$. Additionally, we rank the most to least under-served languages. To illustrate how this ranking helps, let's consider the case for NER in Figure 2. Before the introduction of $Dataset_3$, $L_3$ was most under-served, followed by $L_1$. After inclusion of $Dataset_3$ in the leaderboard, the measured utility for $L_3$ increases. Now, even though the utility value of $L_1$ remains unchanged, $L_1$ might become the most under-served language following the increase in utility of $L_3$. This would act as a positive feedback for the community to direct their efforts towards languages needing most work (here, $L_1$, as specified by the most under-served languages rankings), and drive progress for global equity in a positive *cause-effect* feedback loop.

**Design**: GlobalBench maintains a separate leaderboard for each of the covered tasks, as shown in Figure 1. Each leaderboard details the constituent datasets, system submissions and the following evaluation metrics (further details in §2.2 and §2.3): a) *Performance* (F1, accuracy, BLEU, etc.); b) *System-by-System Utility* (linguistic and demographic); c) *Global Average Utility* (linguistic and demographic); d) *Equity*; e) *Most under-served Languages*; f) *Score by Language*. For more details about GlobalBench UI, please refer to §A.3.

## 2.1 Dataset Selection: *Inclusivity*

The first pillar of GlobalBench's approach is that we attempt to be all-inclusive with respect to tasks, datasets, and languages. On the dataset front, GlobalBench currently includes 966 datasets spanning 190 languages. On the modeling front, it has 1,128 system outputs spanning 6 NLP tasks and 62 languages (note that not every dataset integrated within GlobalBench has system outputs submitted to the leaderboard at present). Overall, GlobalBench has support to accept dataset and system submissions for 17 NLP tasks in 6671 languages[2],

---

[2]The source of metadata of these languages is "Ethnologue: Languages of the World."

managing to include the majority of the about 7000 spoken languages around the world (Austin and Sallabank, 2011). At present, *named entity recognition* is the task with the highest coverage of world speaker population (59.34%), but GlobalBench hopes to continually evolve with time.

## 2.2 Multi-Faceted Evaluation: *Utility and Equity*

**Utility** Blasi et al. (2022) introduce utility $u_l$ of a system for a task and language to be its performance normalized by the best possible performance (typically, human-level performance, but if it's unattainable, we use the empirical maximum as an estimate) afforded by the task:

$$u_l = \frac{performance_l}{theoretical\ max\ performance} \quad (1)$$

While the above helps estimate system performance relative to the ideal scenario, the final utility provided also depends on the systems' *demand*, which is the second term used by Blasi et al. (2022) in their analysis. Demand $d_l$ is characterized by taking into consideration demographic and linguistic perspectives. Under the demographic perspective, the demand for a technology in a language is estimated to be proportional to the number of speakers of the language itself $n_l$ ($d_l \propto n_l$). Under the linguistic perspective, the demand across languages is identical ($d_l \propto 1$). These two alternatives, as well as any intermediate combination of them, are parameterized through a single exponent $\tau$:

$$d_l^{(\tau)} = \frac{n_l^\tau}{\sum_{l' \epsilon L} n_{l'}^\tau} \quad (2)$$

where $\tau = 1$ corresponds to a demographic notion of demand and $\tau = 0$ to a linguistic one. Using the above, Blasi et al. (2022) define a *global metric* as follows:

$$M_\tau = \sum_{l \epsilon L} d_l^{(\tau)} \cdot u_l \quad (3)$$

In essence, $M_\tau = 0$ means that no user benefits from language technology and $M_\tau = 1$ corresponds to each language user enjoying perfect technology.

In GlobalBench, we provide the *demographic weighted* ($\tau = 1$) and the *linguistic weighted* ($\tau = 0$) utilities for all languages in each task. For each language, we take the maximum utility scores

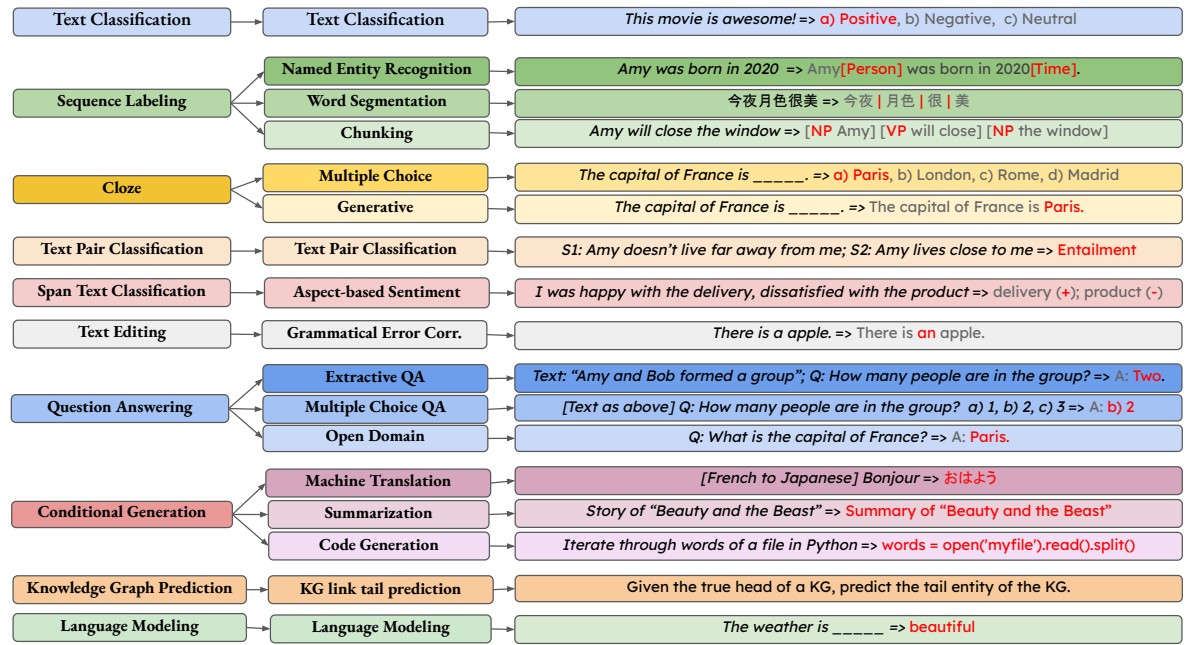

Figure 3: *Overview of Tasks*: All tasks currently supported by GlobalBench are as shown above. This is expected to constantly evolve with time. Refer to Section 4 for details.

of all systems submitted to GlobalBench. To obtain task global averages, as shown in Table 2, we average across utility values for all languages.

**Equity** While utility paints the picture of how far from ideal we are in serving NLP technology to each user, *equity* helps measure how uniform the technologies we serve are, across languages. Khanuja et al. (2023) recently proposed that amongst other measures of statistical dispersion, the Gini coefficient (Dorfman, 1979) best captures this uniformity. Hence, we use the same as a measure of equity in our work. Intuitively, a lower value of G indicates that languages are closer to a uniform distribution. Considering either extremes, when all languages have the same performance, $G = 0$, and when there is support for only one language, $G = 1$. Formally, if performance of a language for a task is $y_i$, ($i = 1 ... n$ where n is the total number of languages), and is indexed in non-decreasing order ($y_i \leq y_{i+1}$), then the Gini coefficient (G) can be calculated as:

$$G = \frac{1}{n} \left( n + 1 - 2 \frac{\sum_{i=1}^{n}(n+1-i)y_i}{\sum_{i=1}^{n} y_i} \right) \quad (4)$$

For each task, we obtain equity values, calculated using the maximum performance of submitted systems for each language in a task. For languages that are not supported by any dataset, we assume

the system performance to be zero. The global equity values for each task are in Table 2.

Apart from the above, we keep track of system performances (F1/Accuracy/BLEU etc.). For each task, we take the system output with the highest performance among all system outputs with the same language, and provide a ranking of languages with the highest system performances. We also maintain a ranking of most under-served languages (sorted in increasing order of utility), for reasons detailed below.

### 2.3 Incentivization: *Reward Improvement*

We can estimate current global progress in language technologies using the demographic and linguistic weighted global averages, and the global equity values. In GlobalBench, we also encourage development of systems that *improve* upon these metrics. We accomplish this in two ways:

First, we identify areas with the greatest potential for improvement, i.e., we identify the *most under-served languages*. Blasi et al. (2022), demonstrates that a variety of values provide different prioritizations over demographic- and linguistic-weighted utility. Following the footsteps of Blasi et al. (2022), we chose a parameter of $\tau = 0.4$ largely qualitatively, observing the results with different values and taking one that seemed to strike a reasonable balance between higher- and lower-

resource languages, such that some of each appeared near the top of the languages to prioritize. Languages farthest from the ideal $\tau$-*weighted utility* are expected to be most under-served. Hence, $(1 - \tau$-*weighted utility*) of a language gives us this measure. We sort each of the 6671 languages supported in GlobalBench according to this measure. In the end, we obtain a ranking of languages with relatively high populations and relatively low scores, broken down by task.

Second, a submission's rank on our leaderboard is determined by how much it contributes to increasing the overall $\tau$-*weighted utility* across languages. This can be achieved in two ways: **i) Data Efforts**: By contributing datasets for previously unsupported languages, their utility, which was previously immeasurable (hence, for all practical purposes, zero), sees a sharp rise; **ii) Improved Systems**: By submitting new systems which improve upon the state-of-the-art, we improve utility by definition (Equation 3).

## 3 Implementation and Participation Details for GlobalBench

GlobalBench accepts open submissions of datasets and systems.

For datasets that are already a part of GlobalBench, system results for them can be submitted to GlobalBench for evaluation. The submission process of system results is simple: one can submit system results to GlobalBench through github directly. Participants don't need to separately submit anything else.

In addition, if people want to submit new datasets to benchmark, one can follow the dataset submission process provided in our github repository.[3] After doing so, corresponding system results can be submitted as per the above instructions.

## 4 Datasets and Tasks

In GlobalBench, we currently support 17 tasks that fall into 10 distinct categories. These tasks represent a diverse set of NLP technologies, ranging from those that are highly applicative and user-facing (question answering, machine translation etc.), to those that aren't directly applied, but are nonetheless fundamental to NLP (language modeling, etc.). We briefly summarize the tasks and provide one example for each task in Figure 3, and we provide a more detailed description and a list of

---

[3] https://github.com/neulab/globalbench.

all datasets that GlobalBench sees system submissions in §A.1. While we support 17 tasks covering 966 datasets, we don't have system outputs for all as of now. We make a note of system outputs available for each task in Table 1, to inform our results and analyses.

## 5 Results and Analysis

GlobalBench allows us to conduct a series of analyses to assess global progress in NLP technologies. While GlobalBench is intended to be evolving to gauge the continued improvements in performance of NLP systems, we examine the current state of systems that have been submitted, to demonstrate how GlobalBench can guide and incentivize future participation and improvement in language technology.

### 5.1 How inclusive is GlobalBench?

The inclusive design of GlobalBench elucidated in Section 2 means that it can (in theory) support any NLP task and dataset. GlobalBench currently supports dataset submissions of 6671 languages and 17 tasks. GlobalBench now covers 966 datasets in 190 languages. We have 1,128 system outputs at the time of writing, spanning 6 NLP tasks: *named entity recognition (NER), text pair classification, text classification, extractive QA, machine translation,* and *KG link tail prediction*; over a total of 62 languages. With the existing systems included in GlobalBench, we already cover 4.72% to 59.34% of the first languages of people in the world depending on the task, as detailed in Table 2. We focus on analyzing system submissions for these six tasks below.

### 5.2 What is our Current Progress as Measured by GlobalBench?

Next, we discuss the current state of NLP progress through the lens of GlobalBench. To do so, we display the demographic- and linguistic-weighted global average scores in Table 2.

**Variance in estimated utility across tasks**: In Table 2, we observe that NER has the highest estimated overall *demographic* and *linguistic* global average. Additionally, NER and MT have the highest language coverage (60 languages). This is because these tasks have been subject to extensive and impressive multilingual dataset development and evaluation efforts by authors of FLORES (Goyal et al., 2022), MasakhaNER (Adelani et al., 2021),

| Category | Task | Number of | | | Metric |
| | | Datasets | Languages | System Outputs | |
|---|---|---|---|---|---|
| Text Classification | Text Classification | 127 | 12 | 199 | Accuracy |
| Sequence Labeling | Named Entity Recognition | 78 | 60 | 450 | F1 |
| | Word Segmentation | 1 | 1 | - | F1 |
| | Chunking | 2 | 1 | - | F1 |
| Cloze | Generative | 10 | 1 | - | CorrectCount |
| | Multiple Choice | 21 | 2 | - | Accuracy |
| Text Pair Classification | Text Pair Classification | 57 | 30 | 96 | Accuracy |
| Span Text Classification | Span Text Classification | 4 | 1 | - | Accuracy |
| Text Editing | Grammatical Error Correction | 10 | 1 | - | SeqCorrectCount |
| Question Answering | Extractive | 80 | 18 | 185 | F1 |
| | Multiple Choice | 72 | 2 | - | Acc. |
| | Open Domain | 4 | 2 | - | ExactMatch |
| Conditional Generation | Machine Translation | 242 | 60 | 170 | Bleu |
| | Summarization | 251 | 55 | - | Bleu |
| | Code Generation | 4 | 5 | - | Bleu |
| KG Prediction | KG Prediction | 3 | 1 | 28 | Hits |
| Language Modeling | Language Modeling | - | - | - | Perplexity |

Table 1: *Overall Statistics* of supported tasks, datasets and system outputs. We currently have 1128 system outputs across 6 tasks and 62 languages. Refer to §5 for a detailed analysis.

and NusaCrowd (Cahyawijaya et al., 2023), which are all included in GlobalBench. In contrast, the estimated *demographic-* and *linguistic-* weighted utility for tasks like KG link prediction or multiple-choice QA are low. These are tasks where intensive data creation efforts have traditionally focused on English. The multilingual datasets that do exist are less widely used and/or not yet included in GlobalBench. However, GlobalBench can help identify these coverage failures and improve accuracy (§5.3).

**Linguistic vs. demographic utility**: In addition, we observe that overall linguistic utility scores across all tasks are very low, and substantially lower than the corresponding demographic utility scores. For instance, the demographic utility score for NER is 0.4489, while the linguistic utility score for the same is only 0.0067. This makes clear that the systems submitted to GlobalBench are currently doing a better job of covering widely-spoken languages, but are doing less well at covering all of the languages in the world.

**Equity of systems across languages**: Since we calculate the Gini coefficient accounting for all 6671 languages, all values are extremely high (nearing 1). Text classification and KG Prediction only have datasets for English, hence the Gini values almost equal 1. We also note that, despite NER

| Task | Demo. Avg. ↑ | Ling. Avg. ↑ | Gini ↓ | % Pop. |
|---|---|---|---|---|
| NER | 0.4489 | 0.0067 | 0.9920 | 59.34% |
| Extractive QA | 0.3460 | 0.0020 | 0.9975 | 44.46% |
| Text Pair Classif. | 0.3465 | 0.0019 | 0.9976 | 39.02% |
| MT | 0.0485 | 0.0002 | 0.9987 | 10.58% |
| Text Classif. | 0.0477 | 0.0001 | 0.9997 | 4.72% |
| KG Prediction | 0.0221 | 0.0001 | 0.9998 | 4.72% |

Table 2: Demographic and linguistic global averages, equity values (Gini), and percentage of world population covered by current submissions of system results to GlobalBench. Only showing tasks with at least one system submission.

and MT having the same language coverage, MT has a higher Gini value, indicating that amongst the languages supported, NER has a more uniform distribution of performance as compared to MT.

**Variation across languages per task**: We also maintain a ranking of system performance for *each language* as described in §2.2. For each task, we take the highest performance amongst all systems, to represent the performance for a (*task, language*) pair. Next, we rank all system performances across languages for each task. For example, suppose we have 4 systems with performances $P_1$, $P_2$, $P_3$, and $P_4$ for task $T_1$, where $P_1$ and $P_2$ are in language $L_1$, and $P_3$ and $P_4$ are in language $L_2$. If $P_1 > P_2$, then $P_1$ is used to represent the performance of $T_1$ in $L_1$; similarly, if $P_3 > P_4$, then $P_3$ is used to represent the performance of $T_1$ in $L_2$. We then rank the

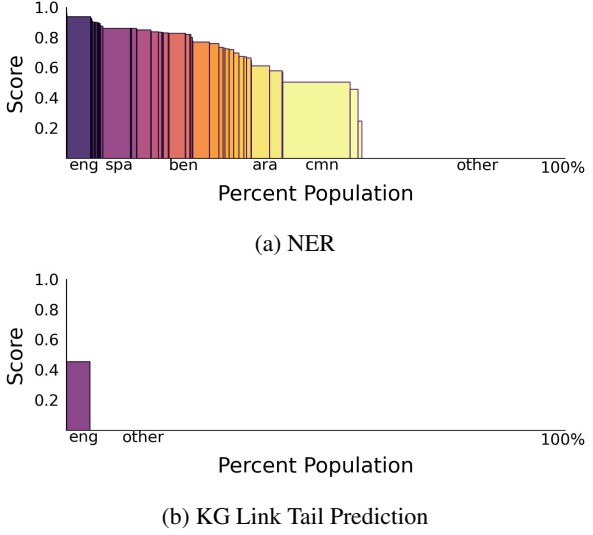

(a) NER

(b) KG Link Tail Prediction

Figure 4: *Variations across languages for a task*: Ranking of system performance for *each language* in NER (above) and KG Prediction (below).

system performances of $L_1$ and $L_2$ under task $T_1$, i.e., we compare $P_1$ and $P_3$ to see which language sees higher system performance. Figure 4a shows a chromatic visualization of system analysis across language populations for NER. GlobalBench supports 78 datasets and 60 languages for this task, and sees 450 system submissions of this task. Therefore, we can see high system performances for many languages. However, KG Link Tail Prediction does not have many submitted systems (28), with monolingual coverage and low performance, as shown in Figure 4b. For a more comprehensive set of chromatic analysis figures for each task with system outputs, please refer to Appendix §A.2.

## 5.3 Measuring improvement with GlobalBench

Another major focus of GlobalBench is to measure and encourage improvement in the quality of language technology systems.

### 5.3.1 How have we improved?

GlobalBench keeps track of when each submission was made, making it possible to examine global progress in NLP over time. To give one example of this, in Figure 5 we show how dataset submissions have helped increase the global averages for NER in the recent past. **[Switch to use publication date of datasets/systems to address reviewer3's question B –YS]** Specifically, the earliest time point only covered English datasets, leaving *both* averages rela-

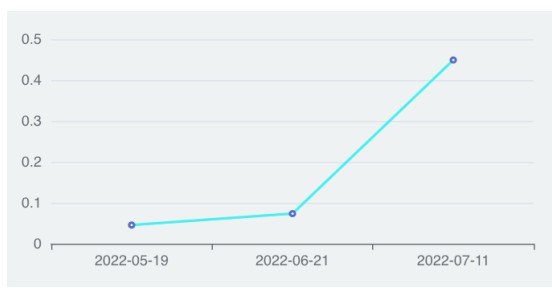

(a) Demographic global average

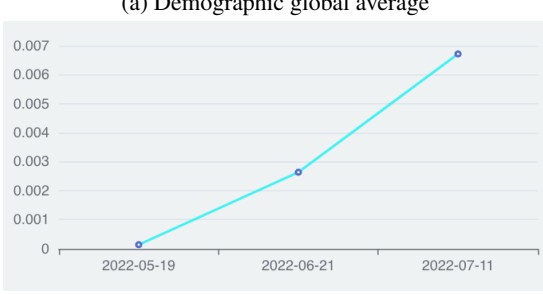

(b) Linguistic global average

Figure 5: A snaphsot capturing the increase in global averages for NER in the recent past, with the addition of new datasets.

tively low. In the second datapoint, systems for African languages from the MasakhaNER dataset were added (Adelani et al., 2021), significantly raising the *linguistic* average. In the third datapoint, systems from the XTREME benchmark (Hu et al., 2020) were added, covering a more populous set of languages, significantly raising the *demographic* average.

### 5.3.2 Where can we improve?

The variety of analysis in the previous section is, in a sense, backward-looking: it looks back at the progress that has *already* been made. In order to instead get a better idea of how we may improve our systems in the future, we use the methodology in Section 2.3 to identify *most under-served languages* in GlobalBench, which are the languages with relatively high populations and relatively low scores for each of the tasks. To display in GlobalBench, we choose a parameter of $\tau = 0.4$, which allows us to moderate between considerations of serving all speakers in the world and serving all languages in the world.

We show the three most under-served languages for each task in Table 3.[4] From these statistics we observe some interesting trends. First, for tasks where the most widely spoken non-English lan-

---

[4]Note that for the purpose of these statistics, we use the source language for the machine translation task.

| Task | Lang 1 | Lang 2 | Lang 3 |
|---|---|---|---|
| Named Entity Recognition | cmn | pnb | wuu |
| Extractive QA | por | jpn | urd |
| Text Pair Classification | ben | por | ind |
| Machine Translation | cmn | spa | ara |
| Text Classification | cmn | spa | ara |
| KG Prediction | cmn | spa | ara |

Table 3: Most under-served languages for each task (by ISO 639-3 language code).

guages in the world, Mandarin Chinese (cmn), Spanish (spa), and Arabic (ara) are not covered, these are selected the most under-served languages. However, for the tasks with better language coverage such as NER, extractive QA, and text pair classification, the most under-served languages are ones with relatively high population that are nonetheless not covered well by existing multilingual datasets that have been included in Global-Bench. This indicates a need for more creation or incorporation of datasets for major languages such as Punjabi, Wu Chinese, and Portuguese, which have been overlooked by existing composite benchmarks.

## 6 Related Work

**From datasets to benchmarks** Given the ubiquitous use of NLP technology in applications, it is imperative to track and maintain progress across a variety of NLP tasks. Evaluating and comparing systems on a single task can also be problematic; past work has identified issues with standard datasets (Artetxe et al., 2020; Gururangan et al., 2018). As the field has progressed, several benchmarks have been released to spur the development of generalizable NLU systems. GLUE (Wang et al., 2018) was one such benchmark with a collection of 9 diverse NLU tasks (sentiment analysis (Socher et al., 2013), natural language inference (Williams et al., 2018), etc.), contrary to prior benchmarks that focused on datasets for a single category of tasks (Conneau and Kiela, 2018). SuperGLUE (Wang et al., 2019) updates GLUE by introducing a new set of harder tasks like commonsense reasoning (Zhang et al., 2018) and question answering (Khashabi et al., 2018). The recently released BIG-bench (Srivastava et al., 2022) consists of a diverse set of 204 tasks, aimed at specifically evaluating the capabilities and limitations of large LMs. Finally, Dynabench (Kiela et al., 2021) is a *human-and-model-in-the-loop* platform, for dynamic data

collection and benchmarking, which currently supports ten tasks. Notably, none of these benchmarks provide utility/equity measures, or a reward structure that incentivizes progress towards the most under-served languages.

**Moving beyond English** While the aforementioned benchmarks have driven progress in NLP for English, there have been several recent efforts made towards other languages as well. Multilingual composite benchmarks such as XTREME (Hu et al., 2020), XTREME-R (Ruder et al., 2021), and XGLUE (Liang et al., 2020) are a collection of datasets in a variety of tasks and languages. XTREME includes 9 tasks across 40 languages, XTREME-R includes 10 tasks across 50 languages with 198 datasets, and X-GLUE improves GLUE (Wang et al., 2018) by including 11 cross-lingual tasks. However, all of these are static and lack the goal to be an all-inclusive, ever-expanding collection of datasets. Beyond these, there have been directed efforts towards dataset curation, especially for the low [text] resource. MasakhaNER (Adelani et al., 2021) supports a large dataset for NER task of 10 African languages. IndoNLU (Wilie et al., 2020) is the first vast resource Indonesian benchmark, and the KLUE benchmark (Park et al., 2021) focuses on 8 diverse tasks in Korean.

In sum, while all of the above efforts have been impactful, none have had the goal to track global progress made by us as a research community, and design a reward system that incentivizes both data and model work, especially for the under-served languages. With GlobalBench, we propose a first step to bridge this gap and move towards inclusive, multi-faceted measurement of progress.

## 7 Conclusion

In this paper, we introduce GlobalBench, an ever-expanding collection of datasets and models spanning *all* tasks and languages within NLP. Our aim is for the community to move towards a *common goal* of building NLP technology that equitably serves all of the world's citizens. To achieve this, we design a leaderboard resting upon three foundational principles: **i)** *inclusivity*: we track progress across all tasks and datasets for 6671 languages; **ii)** *multi-faceted evaluation*: our evaluation measures the per-speaker utility and equity of submitted systems, and also maintains a list of most under-served languages; **iii)** *reward improvement*: we reward data and modeling efforts that help improve utility

across languages, rather than simply maintaining the best-performing systems. Analysing the 1,128 system outputs already integrated within Global-Bench, reveals that NER has the highest utility at present, but is also least equitable. The combination of high demographic and low linguistic utility underscores that efforts have been mostly limited to populous languages. Finally, we identify that the most under-served languages vary across tasks, but are primarily the ones with relatively high speaker population, but nonetheless low coverage in our datasets. All in all, we believe that GlobalBench is one step towards measurable progress in improving the global quality and equity of languages technologies for *all* speakers in the world, and we hope the rest of the research community will join us in pursuit of this goal.

## 8 Limitations

GlobalBench is a broad-reaching effort that has the ambitious goal of measuring performance across all languages in the world. However, to even take the first steps towards this goal we needed to make a number of approximations, which are also inherent limitations of the current work.

**Inclusivity vs. Comparability:** Inclusivity across datasets doesn't come without its downsides. With the goal of covering all datasets, we lose some measure of control to GlobalBench. When evaluating global progress of language technology for particular tasks, GlobalBench uses multilingual datasets that *may* come from distinct sources or have dissimilar genres, causing the difficulty of each dataset to vary. Since GlobalBench doesn't take into consideration the differences in difficulty among datasets of different languages, distinct datasets across different languages might not be directly comparable. However, this is common practice for previous benchmarks such as XTREME (Hu et al., 2020) and Universal Dependencies (Nivre et al., 2016). Furthermore, we expect the law of averages to even out this issue as we keep collecting diverse datasets across domains for each language.

**Languages vs. Language Varieties:** In addition, while GlobalBench relies heavily on distinctions between languages, language boundaries are nebulous, and many dialects or language varieties exist. If datasets annotated with language varieties, as well as demographic information regarding the

number of speakers of these varieties existed, such information could be incorporated within Glob-alBench at a future date. But the current results reported in this paper do not consider this information.

**Reliance on Performance and Population-based Demand Measures:** Currently, GlobalBench relies on standard performance measures such as accuracy, F1, and BLEU to approximate the utility that would be provided by a system to a potential user. However, in reality there is not so direct of a connection between model performance and whether it is actually serving speakers of a particular language well. In addition, we use the first-language speaking population as an approximation for the demand for a language technology in a particular language. However, this disregards second-language speakers, and cannot take into account the case where there may be differing demand for particular pieces of technology by speakers of different languages.

**Possible Bad Actors:** Bad actor is a common concern for benchmarks working on such incentives like GlobalBench. While there may be different varieties of bad actors – those who explicitly cheat, or those who don't explicitly cheat but attempt to get a good score on the benchmark without solving the underlying problem. For the time being, we intend to manually identify and disqualify the former, and we believe that our multi-faceted and broad-spanning approach benchmarking approach may be a sufficient deterrent for the latter.

## Acknowledgements

We gratefully acknowledge support from NSF CAREER Grant No. IIS2142739, the Alfred P. Sloan Foundation Fellowship, and NSF grants No. IIS2125201, IIS2203097, and NSF-FAI 2040926.

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

# A  Appendix

## A.1  Tasks, Datasets and System Outputs

We summarize the tasks and all datasets for which GlobalBench sees submissions of system outputs. The datasets cover a wide range of languages and domains, and vary in terms of size and complexity.

**Text Classification** This task involves classifying text into one or more predefined categories (one label is associated with each sequence). An example of this would be classifying the sentiment of a movie review to be either positive, negative or neutral. GlobalBench covers the following datasets for this task: the QC (Question Classification) dataset (Li and Roth, 2002), the ATIS (Airline Travel Information Systems) dataset (Hemphill et al., 1990), the MR (Movie Review) dataset (Pang and Lee, 2005), the SST-2 (Stanford Sentiment Treebank) Corpus (Socher et al., 2013), datasets from GLUE (the General Language Understanding Evaluation) benchmark (Wang et al., 2018), and the Code-Switching Corpus (Ostapenko et al., 2022).

**Sequence Labeling** This task involves labeling each word or token in a sequence with a specific tag, requiring contextual understanding at the token-level. We cover three tasks here: *Named-Entity-Recognition (NER)*, *Word-Segmentation*, and *Chunking*. *NER* aims to identify and classify named entities, such as person names, organizations, and locations, in a piece of text. *Word-Segmentation* involves identifying the word boundaries of certain languages like Chinese, Japanese, and Thai. *Chunking* involves dividing text into syntactically and semantically coherent chunks or phrases for languages like Chinese. GlobalBench covers the following datasets for this task: the MasakhaNER Corpus (Adelani et al., 2021), the CoNLL-2003 dataset (Tjong Kim Sang and De Meulder, 2003), and the PAN-X dataset (Artetxe and Schwenk, 2019).

**Cloze** This task involves filling in missing words or phrases in a text, based on the context provided by the surrounding words. We focus on *multiple-choice cloze* and *generative cloze*. The former involves filling in the blanks with one of several options, while the latter involves filling in the blanks for a given prompt.

**Text Pair Classification** This task involves classifying the relationship of pairs of texts, such as determining whether two sentences are paraphrases of each other, or contradict each other. A widely known example is that of natural language inference, where the task is to predict whether a given hypothesis can be entailed from or contradicts a premise. GlobalBench covers the following datasets for this task: the Cross-lingual Natural Language Inference (XNLI) corpus (Conneau

et al., 2018), the Stanford Natural Language Inference (SNLI) corpus (Bowman et al., 2015), and the Sentences Involving Compositional Knowldedge (SICK) dataset (Marelli et al., 2014).

**Span Text Classification** This task involves classifying a span of text within a larger piece of text, rather than the whole text. Here, we include *aspect-based sentiment classification* as a task, which involves predicting the sentiment of certain features within a sequence.

**Text Editing** This task involves making corrections or improvements to a piece of text given some requirements, such as fixing grammar or spelling errors. We include the task of *grammatical error correction* here.

**Question Answering** This task involves generating answers to questions either given a context, or in an open-ended fashion. We focus on three tasks: *extractive QA*: extracting the answer span from a given context, *multiple choice QA*: answering a question based on a set of given options, and *open-domain QA*: answering questions from an open domain, such as general knowledge or current events. GlobalBench covers the following datasets for this task: XQuAD (Artetxe et al., 2020), TyDiQA (Clark et al., 2020), SD-QA (Faisal et al., 2021), and MLQA (Lewis et al., 2020).

**Conditional Generation** This task involves generating text based on certain conditions or constraints. We focus on three tasks: *machine translation (MT)*, *summarization*, and *code generation*. *Machine translation* involves translating text from one source language to another where possible MT pairs are counted by considering every possible pair of languages, *summarization* involves generating a concise summary of articles or documents, and *code generation* involves generating a program in a programming language given an input command in natural language. GlobalBench covers the following datasets for this task: datasets from the Fifth Conference on Machine Translation (WMT20) shared tasks[5] and datasets from the Gaokao benchmark (Yuan and Liu, 2022).

**Knowledge Graph Prediction** This task involves reasoning over knowledge graphs (KG), for example, predicting missing or unknown facts in a KG, based on information contained in the graph.

---

[5]https://www.statmt.org/wmt20/

We include the *KG link tail prediction* task, which aims to predict the tail entity of missing links in knowledge graphs. GlobalBench covers the following datasets for this task: WordNet18RR (Shang et al., 2019), FB15K-237 (Bordes et al., 2013).

**Language Modeling**   This task involves predicting the next token in a sequence, given the context of previous tokens. It is a fundamental task used to pre-train decoder models, which are further adapted for downstream applications.

## A.2   Visualization of System Performance across Language populations

We visualize system performances across language populations for each task with at least one system output, as shown in Figure 6.

## A.3   GlobalBench UI

Figure 7 shows the User Interface of GlobalBench Text Pair Classification task. On the top left of the webpage, there is a brief description of the task. Participants will be able to see the statistics of all analyses. For instance, under the Demographic-Weighted Global Average analysis, there is the overall Demographic Average of this task and the diachronic figure representing how the the overall Demographic Average of this task has changed over time.

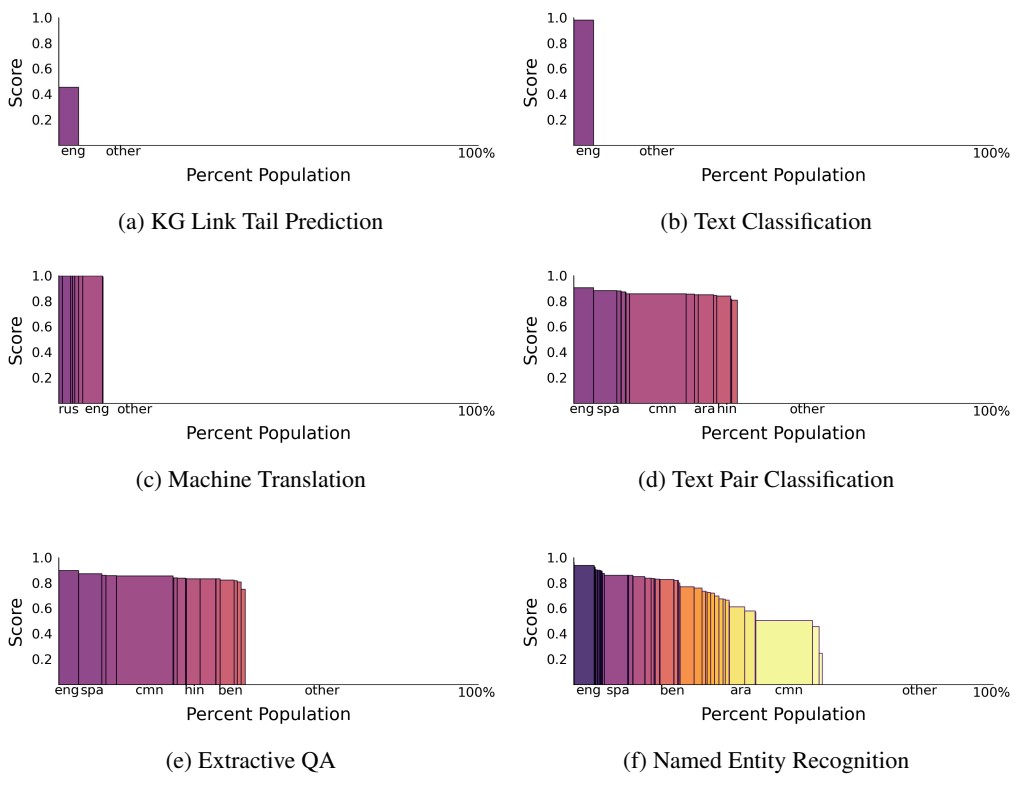

(a) KG Link Tail Prediction

(b) Text Classification

(c) Machine Translation

(d) Text Pair Classification

(e) Extractive QA

(f) Named Entity Recognition

Figure 6: Visualization of System Performance across Language Population

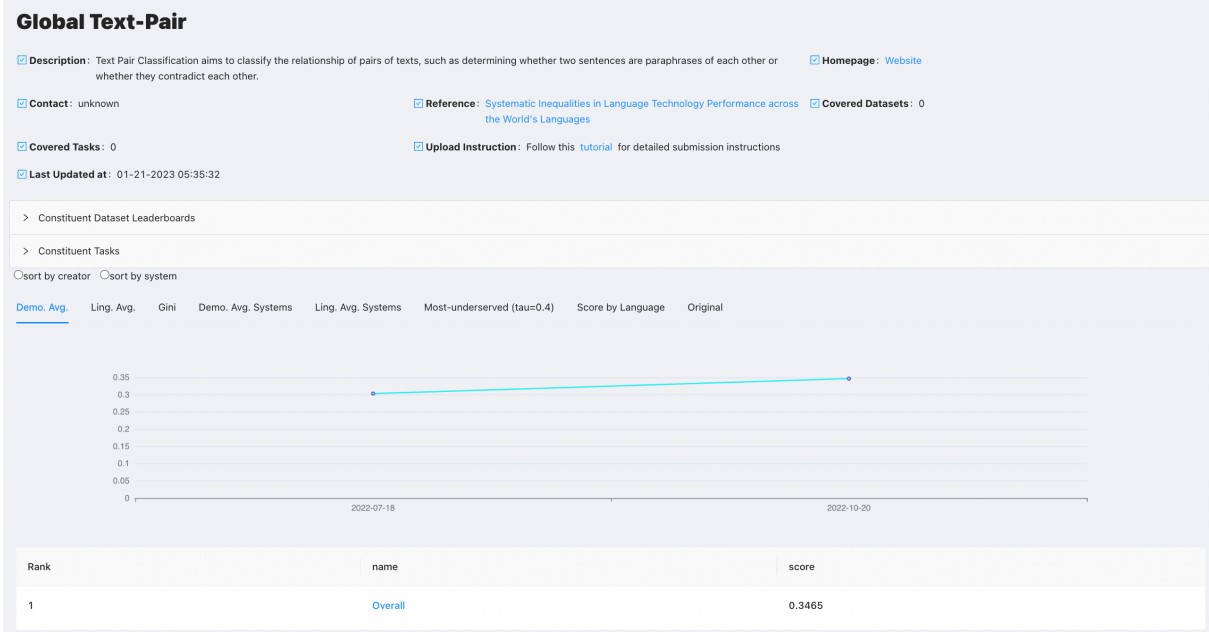

Figure 7: GlobalBench Text Pair Classification task UI