# OpenReview forum: "GlobalBench: A Benchmark for Global Progress in Natural Language Processing"
_EMNLP/2023/Conference — EMNLP 2023 Main_

### Official Review · Reviewer_8Lhb · 2023-08-02

**Soundness:** 4

**Excitement:**

4: Strong: This paper deepens the understanding of some phenomenon or lowers the barriers to an existing research direction.

**Paper Topic And Main Contributions:**

This paper introduces a multilingual benchmark named GlobalBench. Compared to previous works, GlobalBench is:

1) ever-expanding: More datasets can be added in the future.
2) multi-faceted: It evaluates not only the automatic evaluation metrics but also the utility and equity (please see Section 2.2) of both systems and datasets for each language.
3) large: It tries to include "all" datasets. The community is encouraged to upload datasets to this benchmark. Theoretically, it can include all the datasets.

Based on the current version of GlobalBench, the paper analyses the performance disparities among different languages and tasks.

This paper describes an ambitious plan to summarise the current progress of multilingual NLP with a single benchmark, which will require collaborative efforts from the community. The technical contribution of this paper is limited. However, I see this paper as a valuable initiative to identify and develop resources for the most underserved languages.

If the authors are committed to actively maintaining and updating this benchmark for a period of time, this benchmark can be a useful indicator of the status quo of the multilingual NLP. In this case, I look forward to seeing this paper making a meaningful contribution to the community.

**Questions For The Authors:**

Question A: Line 244. How do you choose the parameter value 0.4?

Question B: Section 5.3.1 and Figure 5. The MasakhaNER dataset was added (2021) as the second dataset, and the XTREME dataset (2020) was added as the third dataset. It is very interesting to see how the performance changes over time. Is there a particular order in how these datasets and systems are added to the GlobalBench? Is it ordered by the date the datasets and systems were released? Or is it more random? I am worried that this randomness will bring artefacts to the analysis.

Question C: Table 3. I find it a bit unintuitive that Mandarin Chinese is considered under-served. The research community usually believes Chinese is one of the resource-rich languages. Do you have any explanation for why your analysis shows a different result?

Question D: In line 441. You have mentioned, “spoken non-English languages”. How do you distinguish between spoken and written languages in GlobalBench? Most of the datasets used to train NLP systems are based on written text. How do you determine two spoken languages that may share the same or very similar written scripts? For example, Mandarin Chinese and Wu Chinese?

**Reasons To Accept:**

In addition to automatic evaluation metrics, GlobalBench also evaluates the downstream utility and equity across languages. It is useful to identify languages that may be marginalised or underserved in the NLP research.

It also provides an opportunity to evaluate how a new dataset contributes to more inclusive and equitable language technology, which previous benchmarks are unable to do.

**Reasons To Reject:**

The definitions of utility and equity are all based on automatic evaluation metrics. As mentioned in the Limitation section, it is unclear how these metrics collaborate to the “real” benefit, utility, and satisfaction when users interacting with the system. Alao, directly comparing these automatic metrics across different datasets is also less convincing. However, I acknowledge that how to properly evaluate many NLP systems is still an open research question.

Measuring the progress for a particular task by simply taking the averaged performance or utility of all available datasets is a bit probabilistic, in my opinion. This issue is particularly profound when we are training and evaluating systems on a dynamic set of datasets. For example, introducing a high-quality but difficult dataset will result in a lower total utility value, which is unintuitive. This will not be an issue if we have a fixed set of evaluation dataset.

This benchmark is dynamic, in which new datasets and systems are being added. It is exciting to see how the global utility and equity scores change over time. However, the analysis of this paper is largely based on a single snapshot of the benchmark. Consequently, most of the findings are not very exciting and sound very familiar to previous works. (e.g., https://aclanthology.org/2022.acl-long.376.pdf).

**Reproducibility:**

4: Could mostly reproduce the results, but there may be some variation because of sample variance or minor variations in their interpretation of the protocol or method.

**Reviewer Confidence:**

4: Quite sure. I tried to check the important points carefully. It's unlikely, though conceivable, that I missed something that should affect my ratings.

**Typos Grammar Style And Presentation Improvements:**

Line 2. No need for indentation for the abstract.

Figure 2. I find Figure 2 is not very intuitive. It looks a bit repetitive. For example, in Pillar 2, the top and bottom graphs are almost the same. Pillar 3 also has the same issue. It is also unclear what these dashed arrows ----> are. It would be nice to provide more explanation in the captioning.

Some sentences are a bit repetitive. It would be nice to remove these repetitions. For example, there are many occurrences of “966 datasets”, “17 tasks”, and “a parameter of τ = 0.4”.

---

> ### Author Rebuttal · Authors · 2023-08-29
>
> Thank you for your insightful feedback! We appreciate your recognition of our work as a valuable initiative to identify the most underserved languages and GlobalBench's unique role in assessing contributions of new datasets. We share your excitement for GlobalBench's role as an NLP status indicator and global scores tracker.
>
> We attempt to address the following concerns.
>
> + Regarding the problem of "*how metrics correlate to real benefit*": This is an excellent question, and it is actually on our roadmap for future research work. For a limited number of tasks there are actually research studies demonstrating the relationship between accuracy and user utility (e.g. for translation and speech recognition). For other tasks the relationship is not entirely clear, and thus examining this correlation becomes a first order of business. This is a very interesting topic but due to its scope we, hopefully understandably, would like to pursue it in future work rather than in this paper.
>
> + Regarding the problem of "*directly comparing metrics across different datasets*": As mentioned in the limitations section, we are aware that this is not ideal but most works so far also compare metrics across different datasets.
>
> + Regarding the problem of "*measuring progress of each task by taking the average performance of all datasets*": This is a valid concern, and hence we listed it as a potential limitation. However, as GlobalBench sees more use, we expect multiple datasets being available for each task and language. Aggregating different datasets can help even out discrepancies across domains. In addition, this practice is also established in other benchmarks like XTREME.
>
> + Regarding the problem of "*findings are not very exciting*", we acknowledge that parts of our analysis are based on snapshots, but this is primarily because GlobalBench is a relatively new benchmark and it is still in the process of evolving. We are dedicated to actively managing and updating GlobalBench with new datasets, benchmarks, tasks, and languages. In addition, with the opportunity to present at EMNLP and thus gain greater visibility, we strongly believe that with GlobalBench's continuous expansion, future analysis will reveal more exciting findings and insights such as the changes of global utility and equity scores over time.
>
> + Response to question A ("*How do you choose the parameter value 0.4*"): Blasi et al., which our work is based on demonstrates that a variety of $\tau$ values provide different prioritizations over linguistic and demographic utility. We chose 0.4 largely qualitatively, observing the results with different $\tau$ values and taking one that seemed to strike a reasonable balance between higher- and lower-resource languages (such that some of each appeared near the top of the languages to prioritize). We will explain more about this hyper-parameter choice in the paper.
>
> + Response to question B ("*particular order in how these datasets and systems are added to the GlobalBench*"): This is an excellent point! We are currently using the order of the dataset/system's submission time on GlobalBench in tracking the change in performance over time. We will change this to use the publication date of datasets/systems in the final version.
>
> + Response to question C ("*I find it a bit unintuitive that Mandarin Chinese is considered under-served*"): In measuring most underserved languages, we balance between demographic-weighted (for considerations of serving all speakers) and linguistic-weighted utility (for considerations of serving all languages) as discussed in Section 2.3. Therefore, languages with a lower utility and a larger population would be considered more underserved. In the current nascent incarnation of GlobalBench, there are actually several tasks where there are, as of yet, no Chinese submissions. GlobalBench's population-sensitive utility metrics uncover this, and ideally this will encourage further submissions to fill this current gap.
>
> + Response to question D ("*distinguish between spoken and written languages in GlobalBench*"): This is a valid concern, and hence we listed this problem in the limitations section. GlobalBench depends heavily on distinctions between languages, where language boundaries might be unclear, with many dialects and language varieties, such as the example of Mandarin Chinese and Wu Chinese you mentioned. If we can find relevant information regarding dialects and language varieties, we will include this in GlobalBench. For the current version of GlobalBench, participants submitting datasets are required to identify the language(s) of the submitted dataset.
>
> + Regarding "*presentation improvements*": We will definitely follow your advice of indentation and simplify Figure 2 in the final version. We will also add more explanations to what the arrows are in Figure 2. For the redundant information, we will try to be more succinct and reduce repetitive clarifications in the final version.

---

### Official Review · Reviewer_kyHY · 2023-08-04

**Soundness:** 4

**Excitement:**

4: Strong: This paper deepens the understanding of some phenomenon or lowers the barriers to an existing research direction.

**Paper Topic And Main Contributions:**

This submission describes the design principals and initial tasks and datasets of a multilingual benchmark and leaderboard platform, GlobalBench.
The platform aims to improve the availability of language technologies and datasets across all* of the world's languages, following on from the work of similar multilingual benchmarks. The major contribution of this work is in the fact that it provides specific breakdowns of dataset and technology needs for particular languages, while also providing indications of where developments may have the largest impact in terms of the language community a particular technology would serve.

**Questions For The Authors:**

A. How do the authors propose to ensure or validate the quality of the datasets submitted to the benchmark? Although this is briefly mentioned, this seems like a major risk for the reliability of the benchmarks
B. In addition to the quality, have the authors considered how the size of submitted datasets can be taken into account when calculating utility? Although a new dataset for a language that previously did not have any is surely an improvement, but 10,000 words with NER annotated is certainly not the same as a 300,000 word annotated dataset. This may cause misrepresentations of the utility and equity of languages.
C. Given that the benchmark aims to work on the principal of incentives, have the authors considered how bad actors may manipulate or undermine these incentives, and how their platform will manage these types of submissions?

**Reasons To Accept:**

In general the submission is very well argued and written with a clear delineation of aims, philosophy, and design of the benchmark. The notions of utility also provides a more grounded way of determining what the technology and data needs are for different languages.
This is a very exciting endeavour and I applaud the effort thus far and look forward to see further developments and releases on the platform.


**Reasons To Reject:**

To me there aren't really any clear reasons to reject the submission.
As stated in the questions below addressing the issues of dataset size and quality, as well as considering bad actors in an incentive based platform will further improve the submission.

For ease of reading, the figures and tables should be closer to where they are referenced in the text. E.g. Figure 1 is on p1, while only mentioned on p3, while Fig 2 is already discussed on p2.


**Reproducibility:**

N/A: Doesn't apply, since the paper does not include empirical results.

**Reviewer Confidence:**

4: Quite sure. I tried to check the important points carefully. It's unlikely, though conceivable, that I missed something that should affect my ratings.

---

> ### Author Rebuttal · Authors · 2023-08-29
>
> Thank you for your insightful feedback! We are glad that you find our work well argued with a clear delineation of GlobalBench. We appreciate your acknowledgement that our approach offers a more grounded way to identify the technological and data requirements for various languages.
>
> We attempt to address the following concerns.
>
> + Regarding the problem of "*figures and tables*": We will definitely follow your advice and reformat our figures and tables to make them closer to the text they are referenced in.
>
> + Response to question A ("*How do the authors propose to ensure or validate the quality of the datasets submitted to the benchmark?*"):
> This is a common concern for benchmarks with dynamic datasets, we agree and hence listed it as a potential limitation that we might lose some measure of control. We are considering including a dataset's status of peer-reviewed conferences in the dataset submission process, which may provide some credibility of the dataset. We would very much appreciate any other suggestions on how to mitigate this limitation.
>
> + Response to question B ("*have the authors considered how the size of submitted datasets can be taken into account when calculating utility?*"): This is a very good point and we will certainly think about this more. Note that GlobalBench is not a "leaderboard of datasets" per say, but a leaderboard of systems that also incentivizes dataset creation (as dataset creators will often also be the first to submit a system for that task and language). Because of this, we opt to not explicitly make a judgment on the quality of the submitted datasets, although we may omit datasets that we deem to be of particularly low quality.
>
> + Response to question C ("*how bad actors may manipulate or undermine these incentives and how GlobalBench will manage them*"): This is a valid concern, and we will include this in the limitations section in the final version. Bad actors is a common concern for benchmarks working on such incentives. While there may be different varieties of bad actors – those who explicitly cheat, or those who don't explicitly cheat but attempt to get a good score on the benchmark without solving the underlying problem. For the time being, we intend to manually identify and disqualify the former, and we believe that our multi-faceted and broad-spanning approach benchmarking approach may be a sufficient deterrent for the latter.

---

### Official Review · Reviewer_bbS1 · 2023-08-04

**Soundness:** 3

**Excitement:**

4: Strong: This paper deepens the understanding of some phenomenon or lowers the barriers to an existing research direction.

**Paper Topic And Main Contributions:**

The paper proposes to create a benchmark tool for keeping record of NLP state of the art for many tasks across many languages. The system keeps track of task datasets and submissions to those datasets, calculating metrics for the tasks, and the authors define a set of equations for calculating the utility and equity of the submitted datasets and systems. The approach is interesting in that it would incentivize the creation of datasets for underserved languages and also to improve the state of the art in general. It keeps track of many usual NLP tasks like NER, MT, QA and sentiment analysis, and it could be expanded to include new tasks. However, it seems that so far it has very few datasets for those tasks, e.g. sentiment analysis seems to be covered only for English, although there are datasets for many other languages. It is not clear what language pairs are considered for MT, as it would grow quadratically if we consider all possible combinations. As the platform in its current format relies on using a definition of language that does not count dialects or varieties, it could be a problem as many communities could be left out, this is correctly acknowledged in the limitations section.

**Questions For The Authors:**

- How does the "reward" mechanism work? Are the most important systems listed or highlighted in some way?

- What happens to systems that are submitted but do not beat the current state of the art for the systems? How do you measure if those systems are beneficial in other ways, like being simpler, consuming less resources, etc.

- How are the possible MT pairs counted? Is it only translation from and into English, does it try to consider every possible pair (ideally), or something else?


**Reasons To Accept:**

If it catches on, this platform could help to incentivize both the creation of datasets for underserved languages and the improvement of the state of the art for existing ones.

I liked the idea of taking in consideration the relative number of speakers to measure the impact, but also balancing with a more equitable distribution across languages, trying not to leave languages with few speakers behind.


**Reasons To Reject:**

In a way, this could be seen as an attempt to "gamify" the process of creating datasets and NLP systems, which could attract attention but at the same time could also worsen the problem of racing to create systems that satisfy the benchmarks regardless of their utility in real scenarios. At the same time, according to the definitions in the paper, it seems a system that performs relatively well but does not beat any existing benchmark would not get any recognition, which could be hindering the intention of the system.

It seems that the current version of the platform does not contain many of the common datasets and benchmarks that are used for the different languages. For example, sentiment analysis datasets certainly exist for many languages.


**Reproducibility:**

4: Could mostly reproduce the results, but there may be some variation because of sample variance or minor variations in their interpretation of the protocol or method.

**Reviewer Confidence:**

4: Quite sure. I tried to check the important points carefully. It's unlikely, though conceivable, that I missed something that should affect my ratings.

---

> ### Author Rebuttal · Authors · 2023-08-29
>
> Thank you for your insightful feedback! We are encouraged that you are interested in our incentivization approach. We are grateful that you like our inclusion of the relative number of speakers as a measure of impact while at the same time striking a balance with fair distribution across languages.
>
> We attempt to address the following concerns.
>
> + Regarding the problem of "*racing to create systems that satisfy the benchmarks regardless of their utility in real scenarios*": We fully understand the reviewer's concern, but would argue that this is an issue with all benchmarks, and historically the NLP community has successfully made gradual improvements to adjust to the issues of benchmarks not being aligned with real-world use cases. For example, in question answering, we started with Squad, which evolved into Squad 2.0 adding unanswerable questions, then Natural Questions, which added naturally asked questions, then TyDiQA, which was multilingual. We hope that GlobalBench is a starting point, not an ending point, for development of inclusive NLP benchmarks. In addition, we'd like to highlight that a system can ideally "satisfy our benchmark", only if it contributes to increasing the utility of NLP technology for a language or task. The inclusion of rewarding data contributions is unprecedented and previous benchmarks have only focused on rewarding absolute performance. Hence, we believe that our reward system might better align with utility in the real world. In the future, we'd like to make our reward system multi-dimensional, where systems that are resource-efficient, scalable, and deployable, can also be rewarded.
>
> + Regarding the problem of "*the platform does not contain many of the common datasets and benchmarks*": Currently we have the largest coverage in terms of datasets, tasks and languages (to the best of our knowledge). GlobalBench is also constantly expanding, continually incorporating more datasets, benchmarks, tasks, and languages, with the effort from all participants. With the opportunity to present at EMNLP and thus gain greater visibility, we strongly believe that we can achieve our goal of securing GlobalBench's continuous expansion.
>
> + Regarding "*How does the "reward" mechanism work? Are the most important systems listed or highlighted in some way?*": We maintain separate leaderboards for all systems submitted for each dataset in each task on ExplainaBoard, the benchmarking platform where GlobalBench is built on top of. As shown in figure 7, the first two pages ("demo. avg. and ling. avg.") simply track the average demographic and linguistic utility across all languages for the task. The third leaderboard ("Gini") ranks systems based on how equitable their performance is across all languages supported, and the fourth and fifth ("demo. avg. systems" and "ling. avg. systems"), rank systems based on their demographic and ling utility resp. The "reward" system is implemented through the next page on "most under-served languages". Here, we rank "most under-served languages" by ranking how far the "$\tau$-weighted utility" of language is away from 1 (ideal utility). We are also considering highlighting recent contributions, so that new contributors can also be recognized.
>
> + Response to the question of "*systems that are beneficial in other ways*": This is an excellent idea! We do not have metrics of computational efficiency or other beneficial traits in the leaderboard, but we definitely want to consider ways to incorporate them. One idea might be that we make it possible to view the leaderboard limited to systems that are under a certain parameter size (a similar feature was recently added to the Hugging Face Open LM leaderboard).
>
> + Response to the question of "*MT pairs*": Yes, possible MT pairs are counted by considering every possible pair. Translation is not limited to English: for instance, translation from Indonesian into Turkish is considered a possible MT pair. There are obviously some issues with this – the demographic demand (as calculated and used in section 2.2) for translation may not be equivalent for each pair of languages. We will clarify this in our final version.

---

### Meta-Review · Area_Chair_nWdn · 2023-09-18

**Recommendation:** 5

**Metareview:**

This paper presents GlobalBench: a benchmark that measures global progress in NLP.
GlobalBench improved upon existing multilingual benchmarks such as XTREME or XGLUE. It supports 6671 languages and 17 tasks.
Instead of aggregating accuracy metrics like it's done in most benchmarks, GlobalBench provides a multi-view task-specific dashboard based on three main metrics:
- Performance at a given task (e.g. BLEU for Machine Translation)
- Utility that tracks the downstream utility of a system based on how many speakers can benefit from it and how many languages can benefit from it.
- Equity: Based on the Gini index, it measures how equitable, in terms of language coverage, state-of-the-art NLP technologies is

With Utility, Globalbench provides a leaderboard that incentivizes utility. It ranks all the 6671 supported languages based on a measure of utility that balances demographic and linguistic utility, identifying the languages for which progress is the most needed.
Not only does utility incentivize performance improvement at existing task and dataset, but it also incentivizes the addition of new dataset and new task.

Reasons to accept:
- This paper provides a usable resource to promote measured empirical progress of NLP systems on potentially all languages worldwide (R1, R2 and R3).
- The design choices for Globalbench are novels and thoroughly described
- Specifically, the integration of utility and equity metrics provides a clear incentive to make progress on under-served languages

Reasons to Reject:
- A testable anonymous GlobalBench would have been ideal to make the contribution even more concrete.

---

### Decision · Program_Chairs · 2023-10-07

**Decision:**

Accept-Main

**Comment:**

This paper presents GlobalBench: a benchmark that measures global progress in NLP.
GlobalBench improved upon existing multilingual benchmarks such as XTREME or XGLUE. It supports 6671 languages and 17 tasks.
Instead of aggregating accuracy metrics like it's done in most benchmarks, GlobalBench provides a multi-view task-specific dashboard based on three main metrics:
- Performance at a given task (e.g. BLEU for Machine Translation)
- Utility that tracks the downstream utility of a system based on how many speakers can benefit from it and how many languages can benefit from it.
- Equity: Based on the Gini index, it measures how equitable, in terms of language coverage, state-of-the-art NLP technologies is

With Utility, Globalbench provides a leaderboard that incentivizes utility. It ranks all the 6671 supported languages based on a measure of utility that balances demographic and linguistic utility, identifying the languages for which progress is the most needed.
Not only does utility incentivize performance improvement at existing task and dataset, but it also incentivizes the addition of new dataset and new task.

Reasons to accept:
- This paper provides a usable resource to promote measured empirical progress of NLP systems on potentially all languages worldwide (R1, R2 and R3).
- The design choices for Globalbench are novels and thoroughly described
- Specifically, the integration of utility and equity metrics provides a clear incentive to make progress on under-served languages

Reasons to Reject:
- A testable anonymous GlobalBench would have been ideal to make the contribution even more concrete.